# Therapeutic Effects of Long-Term Administration of Tranilast in an Animal Model for the Treatment of Fibroids

**DOI:** 10.3390/ijms241310465

**Published:** 2023-06-21

**Authors:** Tsai-Der Chuang, Leslie Munoz, Derek Quintanilla, Drake Boos, Omid Khorram

**Affiliations:** 1Department of Obstetrics and Gynecology, Harbor-UCLA Medical Center, Torrance, CA 90502, USA; tchuang@lundquist.org; 2The Lundquist Institute for Biomedical Innovation, Torrance, CA 90502, USA; lesliemunoz1@cdrewu.edu (L.M.); derek.quintanilla@lundquist.org (D.Q.); drake.boos@lundquist.org (D.B.); 3Department of Obstetrics and Gynecology, David Geffen School of Medicine, University of California, Los Angeles, CA 90095, USA

**Keywords:** fibroid, tranilast, xenografts, extracellular matrix

## Abstract

Tranilast (N-3, 4-dimethoxycinnamoyl anthranilic acid) is an orally administered drug with antiallergic properties and approved in Japan and the Republic of Korea for the treatment of asthma and hypertrophic scars. Previous in vitro studies indicated that tranilast reduced fibroid growth through its inhibitory effects on cell proliferation and induction of apoptosis. The objective of this study was to determine the efficacy of tranilast for treatment of human-derived fibroids in a mouse model. SCID mice (ovariectomized, supplemented with estrogen and progesterone) were implanted with fibroid explants and treated for two months with tranilast (50 m/kg/daily) or the vehicle. After sacrifice, xenografts were excised and analyzed. Tranilast was well tolerated without adverse side effects. There was a 37% reduction in tumor weight along with a significant decrease in staining for Ki67, CCND1, and E2F1; a significant increase in nuclear staining for cleaved caspase 3; and reduced staining for TGF-β3 and Masson’s trichrome in the tranilast treated mice. There was a significant inhibition of mRNA and protein expression of fibronectin, COL3A1, CCND1, E2F1, and TGF-β3 in the xenografts from the tranilast-treated mice. These promising therapeutic effects of tranilast warrant additional animal studies and human clinical trials to evaluate its efficacy for treatment of fibroids.

## 1. Introduction

Uterine fibroids are the most common benign fibrotic tumors affecting 40–70% of women during their reproductive years [1,2]. The growth of these tumors is dependent on ovarian steroids, and the tumors are characterized by excess deposition of extracellular matrix (ECM) driven in part by excess expression of pro-fibrotic cytokines such as transforming growth factor-β3 (TGF-β3), platelet-derived growth factor (PDGF), and Activin-A [3,4,5]. The existing medical therapies for fibroids are aimed at inducing a hypoestrogenic state, thereby reducing their growth. These therapies include GnRH (Gonadotropin-releasing hormone) analogues, GnRH antagonists, and progesterone and progesterone receptor modulators that currently are not approved by the US Food and Drug Administration (FDA) [1,2,6]. Due to the hypoestrogenic effects of the GnRH analogues and GnRH receptor antagonist, these drugs can only be used for a limited period. The progesterone receptor modulators have liver toxicity effects along with alteration of endometrial histology, thus limiting their usage [1,2]. Based on the limitations of these existing medical therapies, hysterectomy remains the definitive treatment for these tumors, and there is an urgent need for alternative safe non-hormonal therapies that could be used over extended periods of time without producing adverse effects.

Tranilast (N-3, 4-dimethoxycinnamoyl anthranilic acid) is an orally administered drug with anti-allergic properties and is approved for the treatment of asthma and keloid and hypertrophic scars in Japan and the Republic of Korea [7,8] but currently has no approved indications in the United States or elsewhere. Tranilast inhibits cell growth, the release of inflammatory mediators from mast cells, and collagen biosynthesis, growth factor expression, and TGFβ-induced transformation of fibroblasts into myofibroblast phenotype [8]. Several studies have confirmed the anti-proliferative effect of tranilast in various tumors including prostate, breast, glioma, and pancreatic tumors [7,8]. An early study showed that tranilast inhibits the proliferation of uterine fibroid cells in vitro without inducing apoptosis [9,10]. Tranilast also induced the cyclin-dependent kinase (CDK) inhibitor p21 and the tumor suppressor gene p53 and decreased the expression and activity of CDK2 [9,10,11]. Our group reported that tranilast inhibits leiomyoma smooth muscle cell (LSMC) proliferation by downregulating the expression of cell cycle progression genes cyclin D1 (CCND1), and CDK2. Tranilast also inhibited the expression of collagen Type I (COL1), collagen Type III alpha 1 chain (COL3A1), TGF-β3, DNA methyltransferase 1 (DNMT1), and enhancer of zeste homolog 2 (EZH2) [10] in LSMC. Furthermore, tranilast induced the expression of cellular and secreted miR-29c, which is an anti-fibrotic miRNA targeting many components of the ECM such as collagen [10]. Moreover, we showed that tranilast induces the expression of miR-200c, which targets several cycle-regulatory proteins in fibroids, and this effect is mediated by a transcriptional mechanism involving the inhibition of nuclear factor kappa-B (NF-kB) [11]. Islam et al. also showed an inhibitory effect of tranilast in vitro on fibronectin, COL1A1, versican, and activin A expression in both leiomyoma smooth muscle cells (LSMC) and myometrium smooth muscle cells (MSMC) [12]. Based on the accumulated evidence from in vitro studies indicating a favorable effect of tranilast as an anti-fibrotic and anti-proliferative drug and its inhibitory effect on mast cells, we hypothesized that tranilast could be effective in vivo for the treatment of fibroids by inhibiting cell proliferation and decreasing ECM accumulation, thereby reducing tumor size. This hypothesis was tested in a mouse fibroid model using tumors obtained from hysterectomy specimens.

## 2. Results

### 2.1. Tranilast Has No Side Effects and Is Well Tolerated by Mice

Treatment with tranilast was well tolerated by mice, with no significant effects of tranilast on body weight (vehicle vs. tranilast: 25.7 g ± 0.45 vs. 26.2 g ± 1.03). As shown in Table 1, a blood plasma chemistry panel revealed no adverse effects of in vivo tranilast treatment on carbohydrate metabolism (glucose), kidney function (blood urea nitrogen [BUN], creatinine), electrolytes (sodium, phosphorus), liver function (alkaline phosphatase, hepatocyte glutamic pyruvic transaminase [SGPT; ALT], albumin, total protein, globulin, total bilirubin), or pancreatic amylase.

### 2.2. The Effects of Tranilast on Tumor Size and Expression of Genes Regulating Growth, Apoptosis, and ECM Compsoition

Xenografts from the tranilast-treated group showed a 37% reduction in tumor weight as compared to their matched controls after two months of treatment with tranilast (Figure 1). We then examined the expression of a number of key genes involved in fibroid pathogenesis by qRT-PCR, WB, and ELISA including two major components of the ECM, namely COL3A1 and FN1; two key regulators of the cell cycle (CCND1 and E2F1); and TGF-β3, a critical pro-fibrotic growth factor known to be overexpressed in fibroids [1,10]. As demonstrated in Figure 2, there was a significant decrease in the expression of COL3A1, FN1, CCND1, E2F1, and TGF-β3 mRNA and protein in the xenografts of tranilast-treated mice. We did not find differences in response to tranilast based on race or MED12 mutation status of the tumor. The xenografts were also subjected to histologic and immunohistochemical analysis. Masson’s trichrome staining was used to assess the expression of collagen (blue stain) and smooth muscle cells (red stain) in the xenografts. As shown in Figure 3A,B, there was a significant decrease in the staining for collagen (blue stain) in xenografts of the tranilast-treated mice. In addition, xenografts of tranilast-treated mice exhibited reduced cell proliferation, as indicated by a reduced number of nuclei staining for Ki67, CCND1, E2F1, and TGF-β3 (Figure 3C–J), whereas these xenografts exhibited an increased number of nuclei staining positively for cleaved caspase 3, indicative of increased apoptosis (Figure 3K,L).

## 3. Discussion

This is the first preclinical study examining the efficacy of in vivo administration of tranilast for the treatment of fibroids. We demonstrated that tranilast is well tolerated and shrinks fibroid tumor weight by 37% over the course of two months. Tranilast-induced apoptosis and reduced cell proliferation in fibroid tumors and inhibited ECM deposition, as indicated by reduced staining for collagens in the xenografts. Tumors exposed to tranilast exhibited a favorable gene profile with reduced expression of COL3A1, FN1, CCND1, E2F1, and most importantly, TGF-β3 mRNA and protein. The anti-fibrotic and anti-proliferative effects of tranilast in fibroid tumors in this preclinical study set the stage for larger animal studies with additional doses and duration of treatment and eventually to Phase 1 human clinical trial.

Tranilast exerts its effects through several mechanisms. Prior in vitro studies demonstrated that tranilast inhibits collagen biosynthesis, and this effect is due to its inhibitory effects on TGF-β1 [13]; this was the basis for its use for the treatment of keloids and hypertrophic scars. A recent study provided evidence indicating that tranilast inhibited TGF-β1-induced epithelial–mesenchymal transition (EMT), tumor invasiveness, and metastasis by suppressing Smad4 expression in human non-small cell lung cancer [14]. Our study indicates that tranilast has an inhibitory effect on collagen synthesis in fibroids, and this is most likely due to its inhibitory effect on TGF-β3, which our group and others have shown to be overexpressed in these tumors [1,10]. The inhibition of collagens, which are the main component of the ECM, leads to a reduction in tumor weight as shown here. The 37% reduction in tumor weight after two months of tranilast treatment is similar to the effect of GnRH analogues and, more recently, GnRH receptor antagonists in human clinical trials [2,15]. The inhibitory effect of tranilast on cell proliferation, as indicated by the reduced number of Ki67 positive cells coupled with increased apoptosis, as indicated by the increased number of cells staining for cleaved caspase 3 and inhibition of CCND1 and E2F1, indicates that tranilast is also effective in inhibiting tumor growth and progression. The inhibitory effects of tranilast on cell proliferation in fibroids has also been demonstrated in other cell types [7,8].

Another mechanism of action of tranilast is through stabilization of mast cells and inhibition of histamine release from these cells [16]. Based on this property, tranilast was introduced for the treatment of allergic diseases such as rhinitis, atopic dermatitis, and asthma [16,17]. Mast cells are a known component of fibroid tumors and leiomyosarcoma [18,19,20]. These cells express a number of chemokines such as CCL2, CCL5, CCL11, and TGF-β and possess secretory granules containing heparin, histamine, and serotonin [21]. Heparin was shown to inhibit the proliferation of myometrial and fibroid cells through induction of alpha-SMA, calponin h1, and p27 [21]. Serotonin has been shown to be a growth factor for multiple types of human tumor cells [22] and stimulates collagenase production by uterine smooth muscle via IL1-α [23,24]. A selective blocker of the 5-HT1B receptor inhibited fibroid cell proliferation, induced apoptosis, and reduced cyclin D1 and α-SMA expression [25]. We recently examined serotonin levels in fibroids and matched myometrium but did not find any differences in the levels of serotonin in fibroids compared to its matched myometrium [26]. The role if any of histamine in fibroid pathophysiology remains unknown.

Our group recently reported a marked dysregulation of tryptophan catabolism in fibroids, with upregulation of tryptophan 2,3-dioxygenase (TDO2) and an increase in the breakdown product of this enzyme, kynurenine [26,27]. Another group also reported overexpression of these enzymes in fibroids [28]. Agents that could reduce kynurenine levels could be of potential therapeutic benefit for fibroid treatment. Tranilast is a synthetic derivative of the tryptophan metabolite of tryptophan and a ligand for AhR [29] and could potentially be beneficial for fibroid treatment based on its effect on Trp metabolism. This is evidenced by a case report in which a subject was loaded with tryptophan and then administered 1 g of tranilast daily, which resulted in a marked reduction in urinary kynurenic and quinolinic acid. Based on this finding tranilast was proposed to be a competitive inhibitor of either IDO1 or TDO2 or both [30].

Inflammation plays a significant role in the pathophysiology of fibroids, with dysregulation of multiple proinflammatory cytokines and growth factors [5,31,32]. Our laboratory has demonstrated the significance of NF-κB activation in fibroid pathophysiology, demonstrating increased levels of phosphorylated IKBKB and phosphorylated RelA/p65 [33,34]. Tranilast has an anti-inflammatory effect, and our previous in vitro study demonstrated that the mechanism for this anti-inflammatory effect was through repression of the activity and nuclear translocation of RelA/p65 and the inhibition of binding of RelA/p65 to the miR-200c promoter, thereby inducing miR-200c expression in fibroid cells [11]. The inhibitory effect of tranilast on NF-κB has been shown in other cell types and disease models; in endothelial cells, tranilast inhibited NF-κB-dependent inhibition of adhesion molecules without affecting IκB-α and IκB-ε [35], and in a model of ischemia reperfusion in rats, tranilast blocked the phosphorylation of NF-κB and restored IκB degradation induced by this injury [36].

Tranilast is a well-tolerated drug with a good safety profile for the approved dose of, 300 mg/day for the treatment of asthma [8]. With higher doses of the drug, as used in the large PRESTO clinical trial for the prevention of neointima formation and restenosis after coronary artery dilation, there was an approximate 10% incidence of elevation of liver enzymes, which necessitated early termination of these subjects in the trial [37]. In our study there were no adverse effects of the drug on body weight or blood chemistry. In 3 animals there was a slight elevation of ALT outside of normal range, which was not associated with hyperbilirubinemia or histologic changes in the liver.

The strength of the study is that it is the first preclinical study demonstrating the effectiveness of an anti-inflammatory drug for the treatment of fibroids. The limitation of the study is that detailed mechanistic studies could not be performed due to limited availability of tissue. Other limitations are the limited duration of treatment, which might not reveal the full efficacy of the drug and its potential side effects.

In summary, this preclinical study examining the efficacy of long-term administration of tranilast for the treatment of fibroids revealed a number of beneficial effects, including reduction of tumor weight by 37% over the course of two months. The drug was well tolerated, with only 18% of the animals showing a moderate elevation of ALT probably related to the relative high dose of drug administered. The schematic in Figure 4 demonstrates the potential inhibitory mechanism of action of tranilast on fibroid growth and progression. In this scheme, tranilast inhibits the expression of TGF-β3, which is known to regulate the expression of collagen and fibronectin [3,4,5] and inhibit the expression of cell cycle regulatory proteins (CCND1, E2F1), and increases the level of cleaved caspase 3, resulting in increased cellular apoptosis. Through these effects of tranilast, the volume of the fibroid ECM is decreased. This along with the inhibitory effects of tranilast on cell proliferation and the increase in apoptosis results in decreased tumor size. These results warrant additional studies with different doses of the drug for potentially longer periods of treatment, different forms of delivery, and potentially variants of tranilast that could be more effective. Additional studies will also be needed on fertility and teratogenicity of the drug.

## 4. Materials and Methods

### 4.1. Fibroid Specimens Collection

In order to reduce the variance among fibroids, tumors between 2 cm and 5 cm in diameter and with intramural location (n = 16) were obtained from patients at the Harbor-UCLA Medical Center undergoing hysterectomy. Prior approval from the Institutional Review Board (18CR-31752-01R) at the Lundquist Institute was obtained. Informed consent was obtained from all the patients participating in the study who were not taking any hormonal medications for at least 3 months prior to surgery. The fibroids used in this study were from Caucasian (n = 1), African American (n = 6), Hispanic (n = 8), and Asian (n = 1) women aged 35–60 years (mean 45 ± 6.6 years). The MED12 mutation status of all fibroids was determined by PCR amplification and Sanger sequencing (Laragen Inc., Culver City, CA, USA). Ten tumors were found to have the MED12 mutation. Missense mutations in MED12 exon 2 were the most frequent alteration (n = 9), followed by in-frame insertion–deletion type mutations (n = 1). The missense mutations in exon 2 included c.130G>A (p.Gly44Ser) (n = 4), c.131G>A (p.Gly44Asp) (n = 4), and c.131G>T (p.Gly44Val) (n = 1). The tissues were snap-frozen and stored in liquid nitrogen for further analysis as previously described [40,41].

### 4.2. Fibroid Animal Model

The protocol (31162-02) was approved by the IACUC at the Lundquist Institute at the Harbor-UCLA Medical Center. Female ovariectomized SCID/Beige mice (Charles River Laboratories, Hollister, CA, USA) 9–12 weeks old were implanted with pellets (Innovative Research of America, Sarasota, FL, USA) containing estradiol (0.05 mg/90 d release) and progesterone (25 mg/60 d release) as previously reported [42,43]. A freshly obtained fibroid was cut aseptically into small pieces (4–5-mm^3^ cubes) using a razor blade and weighed. Equal weights of the explants from the same patient were implanted in the flanks of mice, which received tranilast or the vehicle, thus allowing comparison of each treated tumor to its own control. After 3 days of recovery, mice were injected daily intraperitoneal (i.p.) with the vehicle (1% NaHCO_3_) or tranilast (50 mg/kg). The dose of tranilast used was based on a prior publication examining its efficacy in a mouse model of chronic asthma [44]. After two months of treatment animals were sacrificed, and blood was obtained by cardiac puncture. Portions of the livers, kidneys, and hearts were dissected and frozen. Tumor explants were carefully dissected free of surrounding tissue and then were weighed and frozen. Animals were weighed before and after treatment with tranilast or the vehicle.

### 4.3. Histology

After fixation with 4% paraformaldehyde and routine dehydration, a portion of xenograft samples removed after the treatment period ended was embedded in paraffin. To evaluate histologic characteristics, 5-μm-thick paraffin sections of xenografts were stained with hematoxylin and eosin (H-3502, Vector Laboratories, Inc., Newark, CA, USA) and Masson’s trichrome stain (HT15-1KT, Sigma-Aldrich, St. Louis, MO, USA) according to the manufacturers’ instructions.

### 4.4. Safety Test

In order to determine if tranilast administration was safe, we tested the plasma collected from the mice after 8 weeks of treatment with the vehicle or tranilast. The plasma (100 μL) was pipetted into the Comprehensive Diagnostic Profile Rotor (#500-1038, Abaxis, Union City, CA, USA), and the values of glucose, creatinine, BUN, phosphorus, sodium, albumin, alkaline phosphatase, serum glutamic pyruvic transaminase (SGPT; ALT), total protein, globulin, total bilirubin, and amylase were analyzed using the VetScan VS2 chemistry analyzer (Abaxis, Union City, CA, USA).

### 4.5. Quantitative RT-PCR

Total RNA was extracted from the xenografts using TRIzol (Thermo Fisher Scientific Inc., Waltham, MA, USA). RNA concentration and integrity were determined using a Nanodrop 2000c spectrophotometer (Thermo Scientific, Wilmington, DE, USA) and an Agilent 2100 Bioanalyzer (Agilent Technologies, Santa Clara, CA, USA) as previously described [45,46,47]. Briefly, 2 μg of RNA was reverse-transcribed using random primers for selected genes according to the manufacturer’s guidelines (Applied Biosystems, Carlsbad, CA, USA). Quantitative RT-PCR was carried out using the SYBR gene expression master mix (Applied Biosystems). Reactions were incubated for 10 min at 95 °C followed by 40 cycles for 15 s at 95 °C and 1 min at 60 °C. The expression levels of selected genes were quantified using Invitrogen StepOne System, with FBXW2 (F-box and WD repeat domain containing 2) used for normalization [48]. All reactions were run in triplicate, and relative mRNA expression was determined using the comparative cycle treshold method (2^−ΔΔCt^), as recommended by the supplier (Applied Biosystems). Values were expressed as fold change compared to the control group. The primer sequences used were as follows: COL3A1 (sense 5′-ATTATTTTGGCACAACAGGAAGCT-3′, antisense 5′-TCCGCATAGGACTGACCAAGAT-3′), FN1 (sense 5′-ACCGAAATCACAGCCAGTAG-3′, antisense 5′-CCTCCTCACTCAGCTCATATTC-3′), CCND1 (sense 5′-GCCCTCTGTGCCACAGATGT-3′, antisense 5′-CCCCGCTGCCACCAT-3′), E2F1 (sense 5′-GGACTCTTCGGAGAACTTTCAGATC-3′, antisense 5′-GGGCACAGGAAAACATCGA-3′); TGF-β3 (sense 5′-CGGGCTTTGGACACCAATTA-3′, antisense 5′-GGGCGCACACAGCAGTTC-3′), and FBXW2 (sense 5′-CCTCGTCTCTAAACAGTGGAATAA-3′, antisense 5′-GCGTCCTGAACAGAATCATCTA-3′).

### 4.6. Immunoblotting

The total protein isolated from xenografts of the vehicle and tranilast groups was subjected to immunoblotting as previously described [38,39]. Briefly, samples were suspended in a RIPA buffer containing 1 mM EDTA and EGTA (Boston BioProducts, Milford, MA, USA) supplemented with 1 mM PMSF and a complete protease inhibitor mixture (Roche Diagnostics, Indianapolis, IN, USA), sonicated, and centrifuged at 4 °C for 10 min at 14,000 rpm. The concentration of protein was determined using the BCA™ Protein Assay Kit (Thermo Scientific Pierce, Waltham, MA, USA). Equal aliquots (30 micrograms) of total protein for each sample were denatured with SDS-PAGE sample buffer and separated by electrophoresis on an SDS polyacrylamide gel. After transferring the samples to a nitrocellulose membrane, the membrane was blocked with TBS-Tween + 5% milk and probed with the following primary antibodies: COL3A1 and FN1 (Proteintech Group, Inc., Chicago, IL, USA), CCND1 (Cell Signaling Technology, Danvers, MA, USA), and E2F1 (Santa Cruz Biotechnology, Dallas, TX, USA). The membranes were washed with TBS containing 0.1% Tween-20 wash buffer after each antibody incubation cycle. SuperSignal West Pico Chemiluminescent Substrate™ (Thermo Scientific Pierce, Waltham, MA, USA) was used for detection, and photographic emulsion was used to identify the protein bands, which were subsequently quantified by densitometry. The densities of the specific protein bands were determined using the image J program (http://imagej.nih.gov/ij/ (accessed on 10 June 2023)), normalized to a band obtained from staining the membrane with Ponceau S. Results were expressed as means ± SEM as a ratio relative to the control group, designated as 1.

### 4.7. Enzyme-Linked Immunosorbent Assay

The TGF-β3 content in xenografts of vehicle and tranilast groups (n = 16) was measured in duplicate using the Human TGF-β3 ELISA kit PicoKine (Boster Biological Technology, Pleasanton, CA, USA) according to the manufacturer’s instructions. The absorbance of each plate was measured spectrophotometrically at a wavelength of 450 nm, and the concentration was determined by comparing the optical density value of samples with the standard curve. The level of TGF-β3 was calculated as pg/mg of protein and reported as fold change compared to the vehicle group.

### 4.8. Immunohistochemistry

For immunohistochemistry, the xenografts were fixed with 4% paraformaldehyde in PBS and subsequently transferred to PBS containing 30% sucrose (*w*/*v*) until equilibrated in cold (4 °C). After fixation, 5-μm-thick paraffin sections were treated three times with Histo-Clear™ (National Diagnostics, Atlanta, GA, USA) for 5 min, rehydrated by a sequential ethanol wash, and then incubated in a target-retrieval solution (Dako, Carpinteria, CA, USA) in a microwave for 10 min in order to retrieve the antigens. For blocking, tissues were incubated for 10 min with a 3% solution of H_2_O_2_ followed by incubation with PBS-5% normal goat serum-0.2% Triton X–100. Tissue sections were incubated with primary antibody rabbit anti-Ki67 (dilution 1:250, 27309-1-AP, Proteintech Group, Inc, Rosemont, IL, USA), rabbit anticleaved caspase-3 (dilution 1:50, #9664, Cell Signaling Technology, Danvers, MA, USA), rabbit anti-CCND1 (dilution 1:50, #55506, Cell Signaling Technology, Danvers, MA, USA), and mouse anti-E2F1 (dilution 1:50, sc-251, Santa Cruz Biotechnology, Inc., Dallas, TX, USA) overnight at 4 °C in a humidified chamber. The antigens were then visualized using biotinylated antibodies and streptavidin, conjugated with horseradish peroxidase. Control sections were incubated with the secondary antibody, with replacement of the primary antibody with the dilution reagent (Dako). Diaminobenzidine (Dako) served as the substrate. All the sections were counterstained with hematoxylin and eosin. Immunostained sections were examined under a microscope (Olympus IX83; Olympus Surgical Technologies America, San Jose, CA, USA), and five representative images of each slide were quantitatively analyzed with the use of Halo software (Indica Labs Inc., Albuquerque, NM, USA) and were averaged for comparative analysis between vehicle and tranilast treatment groups in a blinded fashion.

### 4.9. Statistics and Power Analysis

Throughout the text, results are presented as mean ± SEM and analyzed by GraphPad PRISM 9 software (Graph-Pad, San Diego, CA, USA). Dataset normality was determined by the Kolmogorov–Smirnoff test. The data presented in this study was not normally distributed, and therefore nonparametric tests were used for data analysis. Comparisons involving two groups was analyzed using the Wilcoxon matched pairs signed rank test as appropriate. Statistical significance was established at *p* < 0.05. Assuming a minimal detectable difference of 25% changes between experimental groups and controls with a 20% expected standard deviation in results, 11 mice are needed at least in each group to achieve a statistical power 0.80 at the level of significance of 0.05 using a two-sided *t*-test.

## Figures and Tables

**Figure 1 ijms-24-10465-f001:**
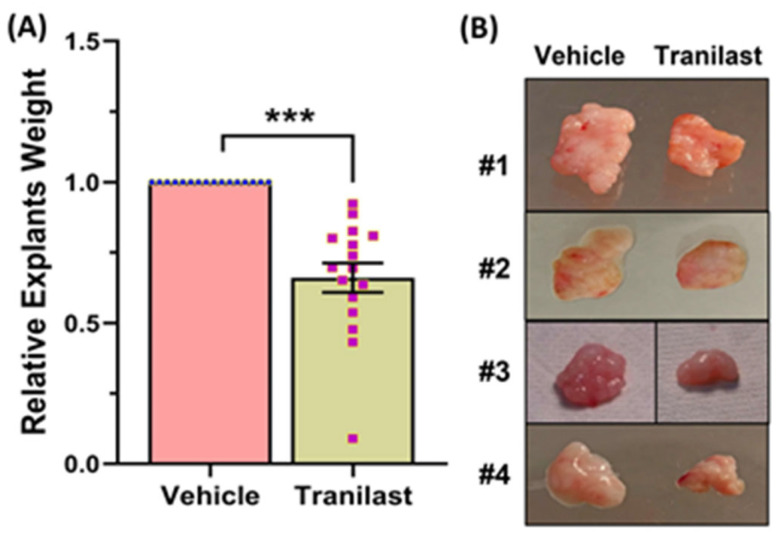
(**A**) Fresh fibroids explants were implanted subcutaneously in the ovariectomized CB-17 SCID/Beige mice, and the vehicle or tranilast (50 mg/kg/daily) was administered i.p. daily for 8 weeks. The weight of tumor explants was determined after 8 weeks of treatment (n = 16). (**B**) Representative images of four xenografts at the end of treatment period (8 weeks). The results are presented as mean ± SEM of independent experiments, with *p* values indicated at the corresponding line. *** *p* < 0.001.

**Figure 2 ijms-24-10465-f002:**
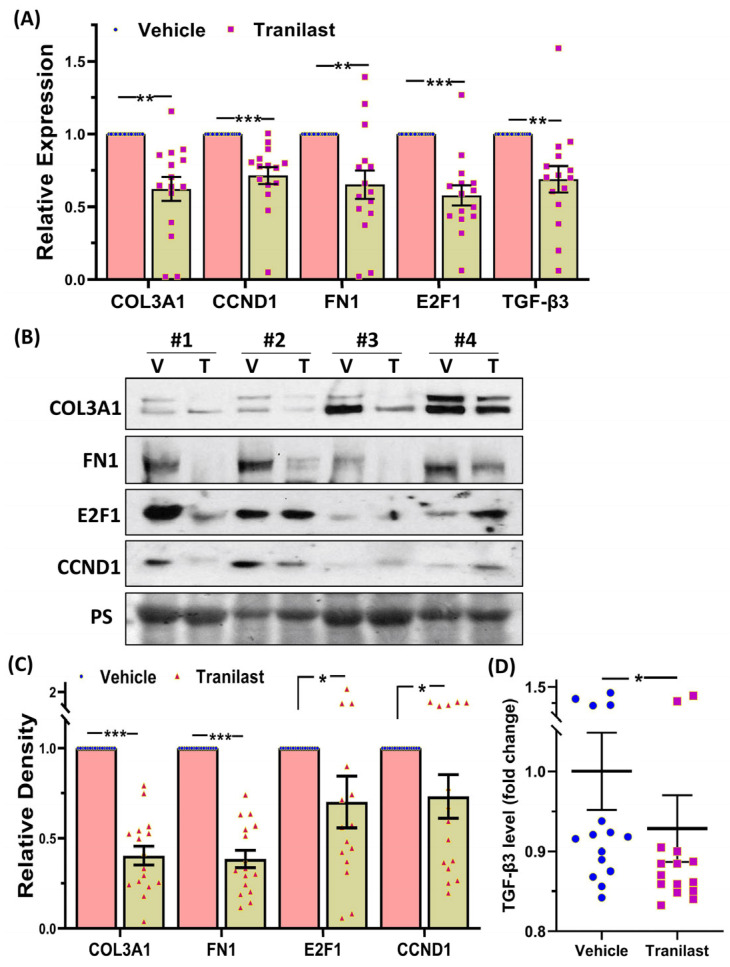
(**A**) Relative expression of COL3A1, CCND1, FN1, E2F1, and TGF-β3 mRNA in xenografts implanted subcutaneously in the ovariectomized CB-17 SCID/Beige mice (n = 16) following 8 weeks of treatment with the vehicle or tranilast (50 mg/kg/daily). (**B**) Representative Western blot analysis of COL3A1, CCND1, FN1, and E2F1 with bar graphs (**C**) showing their relative band densities in the xenografts (n = 16). (**D**) TGF-β3 levels as determined by enzyme-linked immunosorbent assay in 16 xenografts. The results are presented as mean ± SEM, with *p* values indicated by corresponding lines. * *p* < 0.05; ** *p* < 0.01; *** *p* < 0.001.

**Figure 3 ijms-24-10465-f003:**
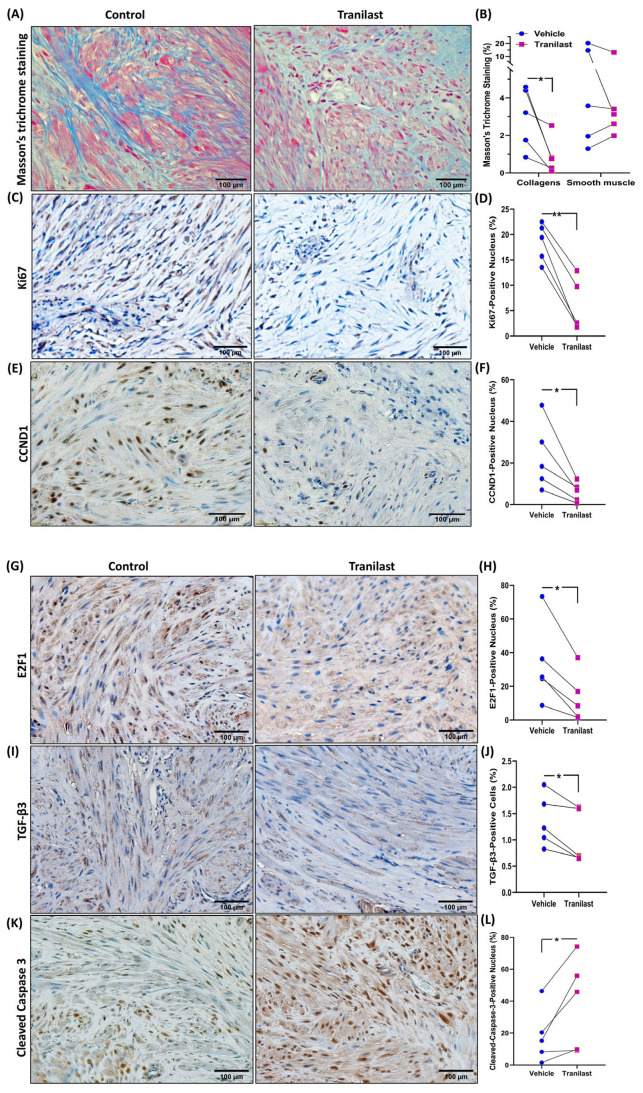
(**A**,**B**) Representative histopathological images determined by Masson’s trichrome staining of fibroid xenografts from vehicle or tranilast-treated group (magnification, ×20). Blue color demonstrates collagen fibers, and red color indicates smooth muscle cells. (**B**) Shows the quantification of staining intensity by Halo software (Indica Labs-Area Quantification v2.4.2) (n = 5 in each group). (**C**–**L**) Representative immunohistochemically stained images of fibroid xenografts treated with vehicle or tranilast (magnification, ×20) for Ki67 (**C**,**D**), CCND1 (**E**,**F**), E2F1 (**G**,**H**), and TGF-β3 (**I**,**J**) and cleaved caspase 3 (**K**,**L**). Quantification analysis by Halo software (Indica Labs-Multiplex IHC v3.2.3) for the positive staining in nucleus (**D**,**F**,**H**,**L**) and cytosol (**J**) (n = 5 in each group). The results are presented as mean ± SEM, with *p* values indicated at the corresponding line. * *p* < 0.05; ** *p* < 0.01.

**Figure 4 ijms-24-10465-f004:**
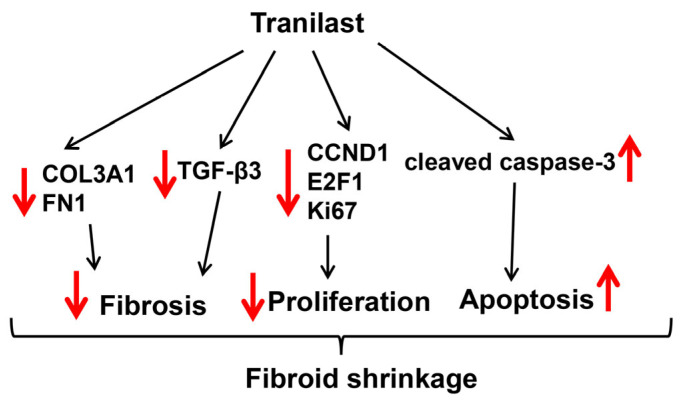
Schematic diagram representing the potential inhibitory mechanism of tranilast on fibroid growth and progression. Fibroids are characterized by increased expression levels of COL3A1, FN1, and TGF-β3 (ECM deposition and tissue fibrosis) [10,38] and CCND1 and E2F1 (mediated cell proliferation) [10,39]. Tranilast treatment significantly inhibits the expression of these genes while inducing apoptosis through increased levels of cleaved caspase 3, resulting in repressed ECM deposition, proliferation, and decreased fibroid size.

**Table 1 ijms-24-10465-t001:** The Effects of Tranilast on Blood Chemistry of Mice.

Chemistry Panel Marker	Vehicle	Tranilast	*p*-Value
	(Mean ± SEM)	(Mean ± SEM)	
General metabolism			
Glucose (mg/dL)	228.8 ± 13.12	215.4 ± 13.08	0.47
Kidney function			
BUN (mg/dL)	23.5 ± 2.40	23.7 ± 2.54	0.18
Creatinine (mg/dL)	0.2 ± 0.01	0.2 ± 0.01	0.43
Electrolytes			
Sodium (mEq/L)	153.9 ± 0.72	155.5 ± 1.43	0.33
Phosphorus (mg/dL)	8.6 ± 0.49	8.7 ± 0.58	0.72
Liver function			
Alkaline phosphatase (U/L)	90 ± 13.49	94 ± 13.53	0.68
Albumin (g/dL)	3.7 ± 0.12	3.5 ± 0.16	0.3
SGPT (ALT) (U/L)	54.2 ± 10.18	112.7 ± 50.87	0.27
Total protein (g/dL)	4.9 ± 0.41	4.5 ± 0.11	0.26
Globulin (g/dL)	1.2 ± 0.34	1.0 ± 0.12	0.48
Total bilirubin (mg/dL)	0.2 ± 0.02	0.2 ± 0.02	0.72
Pancreas function			
Amylase (U/L)	692.0 ± 22.47	725.0 ± 90	0.77

## Data Availability

Raw data were generated at the Lundquist Institute. Derived data supporting the findings of this study are available from the corresponding author O.K. on request.

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
