# Peer review of "Therapeutic Effects of Long-Term Administration of Tranilast in an Animal Model for the Treatment of Fibroids"

_ijms, 2023, doi:10.3390/ijms241310465_

Round 1

Reviewer 1 Report

The study is very interesting and is part of that set of clinical studies to try to find an alternative pharmacological remedy in the treatment of fibroids.

There are some passages that are not clear to me in the M&Ms.

In the first part, Fibroid Specimens Collection, the paragraph should be better rephrased.

Did women first undergo hysterectomy or myomectomy alone, in order to get the fibroid to implant in mice?

The age of the women enlisted is from 35 to 60 years old.

A woman over 50 does not do myomectomies, but she does, usually, hysterectomy.

The diameter of the fibroids was estimated to be 2.5 cm. Why not bigger? The women were of various races, which is a limitation of the study.

It would be useful to include, at the end of the discussion, the strengths and limitations of the study.

It is unclear whether fibroids implanted in mice were first tested with hematoxylin and eosin before implantation or only afterwards.

Why only two months of tranilast and not the classic 3 months (90 days)?

Author Response

  1. The study is very interesting and is part of that set of clinical studies to try to find an alternative pharmacological remedy in the treatment of fibroids. There are some passages that are not clear to me in the M&Ms. In the first part, Fibroid Specimens Collection, the paragraph should be better rephrased. Did women first undergo hysterectomy or myomectomy alone, in order to get the fibroid to implant in mice? The age of the women enlisted is from 35 to 60 years old. A woman over 50 does not do myomectomies, but she does, usually, hysterectomy. The diameter of the fibroids was estimated to be 2.5 cm. Why not bigger? The women were of various races, which is a limitation of the study.

Response: Thanks for the suggestions. We have revised the fibroid specimen collection section in the M&M as below. The specimens were from hysterectomy specimens, and we used fibroid between 2 to 5 cm in diameter not 2.5 cm in diameter. We have removed the dash and used the word between ‘2 and 5 cm’ to eliminate the confusion. The reason for limiting the size parameter for fibroids was to reduce the heterogeneity and variance among tissues. This rationale has been included in the paper.

Line 240-242: “In order to reduce the variance among fibroids, tumors between 2 to 5 cm in diameter and with intramural location (n=16) were obtained from patients at Harbor-UCLA Medical Center undergoing hysterectomy.”

  1. It would be useful to include, at the end of the discussion, the strengths and limitations of the study.

Response: Thanks for the suggestions. We have included a paragraph on strength and limitations.

Line 217-221: “The strength of the study is that it is the first preclinical study demonstrating the effectiveness of anti-inflammatory drug for the treatment of fibroids. The limitation of the study is that detailed mechanistic studies could not be performed due to limited availability of tissue. Other limitations are the limited duration of treatment which might not reveal the full efficacy of the drug and its potential side effects.”

  1. It is unclear whether fibroids implanted in mice were first tested with hematoxylin and eosin before implantation or only afterwards.

Response: Thanks for the suggestions. Hematoxylin staining was done on the xenografts following the treatment period. This has been clarified in the M&M section as follow:

Line 271-272: “After fixation with 4% paraformaldehyde and routine dehydration, a portion of xenograft samples removed after the treatment period ended were embedded in paraffin.”

  1. Why only two months of tranilast and not the classic 3 months (90 days)?

Response: Thanks for the suggestions. This was a pilot study to determine if tranilast is effective in a short period of time. Longer duration studies are planned for the future and will potentially produce more profound effects on fibroid growth and progression.

Reviewer 2 Report

In the article titled “Therapeutic Effects of Long-term Administration of Tranilast In An Animal Model for the Treatment of Fibroids” the authors determine the efficacy of tranilast for treatment of human derived fibroids in a mouse model. They demonstrated that tranilast is well tolerated and shrinks fibroid tumor weight by 37% over the course of two months. Tranilast induced apoptosis and reduced cell proliferation in fibroid tumors and inhibited ECM deposition. Tumors exposed to tranilast exhibited a favorable gene profile with reduced expression of COL3A1, FN1, CCND1, E2F1 and  TGF-β3 mRNA and protein.

 I think that I can reconsider the possibility of publication after a major revision.

My suggestions are the following:

 In the sentence “Previous in vitro studies indicated that tranilast inhibited fibroids growth through its inhibitory effects on cell proliferation and induction of apoptosis” lines 15-16 change one of the two word: inhibited or inhibitory with synonyms

The sentence in the lines 70-75 is too long.

 Tables should have a short explanatory title and caption. It is not clear that of table 1

 enlarge table 1 a bit

Why does Col3A1 show 2 bands in western blotting?

 Can you provide a better image replica of western blotting? For some proteins the results are not very clear and consequently the quantifications

 Explain the molecular mechanism of action of Tranilast emerging from this trial and make a model.

I believe that to make the article more appealing you should include a nice color image that summarizes the results, the molecular mechanism and the message the authors want to give

 What are the limitations of this study?

Moderate editing of English language required

Author Response

  1. In the sentence “Previous in vitro studies indicated that tranilast inhibited fibroids growth through its inhibitory effects on cell proliferation and induction of apoptosis” lines 15-16 change one of the two word: inhibited or inhibitory with synonyms.

Response: Thanks for the suggestions. This sentence was revised as below.

Line 15-16: “Previous in vitro studies indicated that tranilast reduced fibroids growth through its inhibitory effects on cell proliferation and induction of apoptosis.”

  1. The sentence in the lines 70-75 is too long.

Response: Thanks for the suggestions. This sentence was revised as below.

Line 70-72: “Based on the accumulated evidence from in vitro studies indicating a favorable effect of tranilast as an anti-fibrotic and anti-proliferative drug and its inhibitory effect on mast cells [13], we hypothesized that tranilast could be effective in vivo for the treatment of fibroids by inhibiting cell proliferation and decreasing ECM accumulation thereby reducing tumor size.”

  1. 3. Tables should have a short explanatory title and caption. It is not clear that of table 1. Enlarge table 1 a bit

Response: Thanks for the suggestions. Captions have been included and the table 1 was enlarged.

  1. Why does Col3A1 show 2 bands in western blotting?

Response: Thanks for the suggestions. The Col3A1 antibody used in this study consistently identified two bands one of which might be procollagen. We have included both bands in the density determination.

  1. Can you provide a better image replica of western blotting? For some proteins the results are not very clear and consequently the quantifications

Response: Thanks for the suggestions. We have replaced with better representative western blotting image.

  1. Explain the molecular mechanism of action of Tranilast emerging from this trial and make a model. I believe that to make the article more appealing you should include a nice color image that summarizes the results, the molecular mechanism and the message the authors want to give

Response: Thanks for the suggestions. We have included a paragraph and figure 4 to summarize the results.

Line 226-233: “The schematic in figure 4 demonstrates the  potential inhibitory mechanism of action of tranilast on fibroid growth and progression. In this scheme, tranilast inhibits the ex-pression of TGF-β3, which is known to regulate the expression of collagen and fibronectin [3-5] and inhibits the expression of cell cycle regulatory proteins (CCND1, E2F1), and increases the level of cleaved caspase 3 resulting in increased cellular apoptosis. Through these effects of tranilast the volume of the fibroid ECM is decreased. This along with the inhibitory effects of tranilast on cell proliferation and increase in apoptosis result in decreased tumor size.”

  1. What are the limitations of this study?

Response: Thanks for the suggestions. We have included a paragraph on strength and limitations.

Line 217-221: “The strength of the study is that it is the first preclinical study demonstrating the effectiveness of anti-inflammatory drug for the treatment of fibroids. The limitation of the study is that detailed mechanistic studies could not be performed due to limited availability of tissue. Other limitations are the limited duration of treatment which might not reveal the full efficacy of the drug and its potential side effects.”

Reviewer 3 Report

It is an interesting paper on the reduction of fibroids in a mouse model.

Better structure, more precise reporting and sample size calculation are the issues to be revisited.

adequate 

Author Response

  1. Better structure, more precise reporting and sample size calculation are the issues to be revisited.

Response: Thank you for the kind comments. We included a paragraph on Power calculation in the M&M section.

Line 363-366: “Assuming a minimal detectable difference of 25% changes between experimental groups and controls, with a 20% expected standard deviation in results, 11 mice are needed at least in each group, to achieve a statistical power 0.80 at the level of signifi-cance of 0.05 using a two-sided t-test.”

Round 2

Reviewer 2 Report

The authors answered all my questions. I accept the manuscript in the present form

Minor editing of English language required